# Assessing the Antimicrobial Properties of Honey Protein Components through In Silico Comparative Peptide Composition and Distribution Analysis

**DOI:** 10.3390/antibiotics12050830

**Published:** 2023-04-28

**Authors:** Andrzej Łyskowski, Michał Miłek, Małgorzata Dżugan

**Affiliations:** 1Faculty of Chemistry, Rzeszów University of Technology, Powstańców Warszawy 6, 35-959 Rzeszów, Poland; 2Department of Chemistry and Food Toxicology, Institute of Food Technology and Nutrition, University of Rzeszów, Ćwiklińskiej 1a, 35-601 Rzeszów, Poland

**Keywords:** *A. cerana*, *A. mellifera*, antimicrobial peptides, honeybee, in silico, proteome

## Abstract

The availability of reference proteomes for two honeybee species (*Apis mellifera* and *Apis cerana cerana*) opens the possibility of in silico studies of diverse properties of the selected protein fractions. The antimicrobial activity of honey is well established and related to its composition, including protein components. We have performed a comparative study on a selected fraction of the honey-related proteins, as well as other bee-secreted proteins, utilizing a publicly available database of established and verified peptides with antimicrobial properties. Using a high-performance sequence aligner (diamond), protein components with antimicrobial peptide sequences were identified and analyzed. The identified peptides were mapped on the available bee proteome sequences, as well as on model structures provided by the AlphaFold project. The results indicate a highly conserved localization of the identified sequences within a limited number of the protein components. Putative antimicrobial fragments also show high sequence-based similarity to the multiple peptides contained in the reference databases. For the 2 databases used, the lowest calculated percentage of similarity ranged from 30.1% to 32.9%, with a respective average of 88.5% and 79.3% for the *Apis mellifera* proteome. It was revealed that the antimicrobial peptides (AMPs) site is a single, well-defined domain with potentially conserved structural features. In the case of the examples studied in detail, the structural domain takes the form of the two β-sheets, stabilized by α-helices in one case, and a six-β-sheet-only domain localized in the C-terminal part of the sequence, respectively. Moreover, no significant differences were found in the composition of the antibacterial fraction of peptides that were identified in the proteomes of both species.

## 1. Introduction

The antimicrobial activity of honey is a well-known characteristic that underpins its health benefits. For centuries, honey has been used in therapy, e.g., in the treatment of difficult-to-heal wounds, although the mechanisms of action on pathogenic microorganisms have been discovered relatively recently [1]. There are several mechanisms of antimicrobial action in compounds of honey, related to the presence of low-molecular-weight compounds (e.g., methylglyoxal and polyphenols) and high osmotic pressure resulting from the high sugar content, low water activity, and low pH value (Table 1) [1,2,3]. One of the most significant features is the generation of hydrogen peroxide, as a result of the activity of the glucose oxidase enzyme [4,5,6]. It is an enzyme of bee origin that catalyzes glucose oxidation to gluconic acid, and simultaneously generates H_2_O_2_, producing an antimicrobial effect [6]. Another important factor is the presence of certain proteins and peptides that affect microorganisms. The most important are major royal jelly proteins (MRJPs), for which antimicrobial, anti-inflammatory, and anticancer activities have been demonstrated [7]. MRJP1 is a precursor to three antimicrobial entities, collectively called jelleins [8,9]. Moreover, one of the main factors responsible for the effect of honey on microorganisms is the defensin-1 peptide, also known as royalisin. It is produced by bees in the hypopharyngeal glands, from which it is then introduced into the honey and royal jelly they produce [2,10].

Honey shows a wide spectrum of beneficial properties, including antimicrobial action, which strongly depends on botanical (variety of honey) and geographical origins [21]. It has been shown that honeys of darker varieties (e.g., buckwheat, heather, and honeydew) contain more bioactive compounds and have stronger antioxidant and antimicrobial effects [13,22]. Geographical origin may, in turn, be related to the involvement of other bee species in the production of honey. In studies on the physicochemical and antioxidant properties of honeys produced in Malaysia by *Apis mellifera, A. dorsata*, and *A. cerana*, the strongest antioxidant effect (usually associated with antimicrobial activity) was shown by honeys produced by *A. cerana* [23]. Similarly, another bee product, propolis, varies significantly, depending on the bee species producing it [24,25].

The antimicrobial action of natural products is achieved primarily through the action of small molecules that interfere with various basic life-sustaining processes that lead to, at least, the suppression of the growth of harmful microorganisms such as bacteria, viruses, or fungi [26]. Such agents can be classified, according to their mode of action, as microbicides if they lead to the death of the microorganism, or as bacteriostatic agents if they only prevent the uncontrolled growth of the potential pathogen [27]. Numerous mechanisms of action (Table 1) make honey an effective agent against various dangerous pathogens, such as antibiotic-resistant bacteria (e.g., MRSA, *Shigella* sp., *Listeria monocytogenes*, *Helicobacter pylori*) [3,28]. A relatively new class of chemicals, investigated in the treatment and prevention of bacterial infections, is antimicrobial peptides (AMPs). They encompass a class of small peptides that are widely distributed in nature, and can play an important role in the responses of the innate immune system of various organisms [29,30,31]. The physicochemical properties, structures, and modes of action differ extensively in this group. Unlike most small-molecule antibiotics, AMPs exhibit a wide range of inhibitory effects against their targets, and act in a more generalized mechanism. Additionally, AMPs tend to act on multiple targets, making the development of resistance and the parallel transfer of resistance more difficult for microorganisms [27,30].

Due to such a widely distributed effect and mode of action, a reliable prediction of the AMPs has proven to be a challenge. Multiple strategies have been employed with different rates of success [32,33,34]. At the same time, numerous collections of antimicrobial peptides have been created [35,36]. The Antimicrobial Peptide Database (APD3, https://aps.unmc.edu/), at the University of Nebraska Medical Centre, is a representative example. The APD3 contains over 3500 sequences (as of January 2023) from all 6 life kingdoms, the most dominant group coming from animals [37]. The collection contains naturally occurring and gene-predicted, as well as some synthetic, peptides. The supporting information collected, associated with the entries, includes data on the source organism, associated peptide family name, life domain, biological activity, target microbes, known molecular targets and synergistic effects, mechanism of action, structural information, availability, and bibliographic information. The classification of the biological mode of action contains more than 25 categories. The Database of Antimicrobial Activity and Structure of Peptides (DBAASP, https://dbaasp.org/ accessed on 23 January 2023) is another example of the available resources, with over 19 000 entries, and equally impressive and complete background information for collected data [38]. Both databases make the collection freely available to researchers [36].

The in silico evaluation of the antimicrobial properties of honey would not be possible without complete proteome information on the respective bee species. There are currently two reference proteomes available from the UniProt proteome collections: *Apis mellifera* (honeybee, proteome ID: UP000005203) and *Apis cerana cerana* (oriental honeybee, proteome ID: UP000242457) [39]. The first comprises 19 054 proteins and is reported to be 98.9% complete; the second comprises 9 931 proteins and is 91.3% complete.

With both sets of data at hand, we have performed an in-depth comparative in silico proteome antimicrobial peptide composition and distribution analysis. With the help of the data generated by the AlphaFold Protein Structure Database initiative (https://alphafold.com/ accessed on 23 January 2023), the generated data were filtered and mapped on prospective proteomic honey components, allowing us to verify predictions on the structural level [40].

## 2. Results

### 2.1. Evaluation of Reference AMPs Databases

Reference AMP databases selected for analysis contain a significant amount of information. However, because the basis of this information is an amino acid sequence, we have decided to investigate the level of redundancy of the contained information. For the purpose of this analysis, we have designated the APD3 database as a reference. When the content of the APD3 database is aligned against itself, a clear presence of non-unique sequences is visible (Figure 1a). The percentage identity distribution of the aligned peptide sequences indicates the dominant fraction with a 100% identity, which is a clear indication of the unique content of the database. It is also clear that about half of the obtained hits have significant similarity (more than 60%) in their amino acid sequence composition. When the two databases are compared against each other, and the content of the DBAASP database is aligned with the APD3 database, a similar trend is observed (Figure 1b). However, a dominant peak at 100% indicates that both databases contain a significant portion of overlapping AMP sequences. The distribution of similar sequences is also shifted toward a higher percentage of sequence identity score.

### 2.2. Distribution of AMPs in the Analyzed Proteomes

Proteomes of the *A. mellifera* and *A. cerana cerana* were analyzed by aligning the content of the reference AMP database’s content. A graphic representation of the results obtained is presented in the figures below: in Figure 2 for *A. mellifera*, and in Figure 3 for *A. cerana cerana*. The alignment procedure yielded 121 and 907 hits for *A. mellifera* analysis with APD3 and DBAASP, respectively, and 199 and 516 hits for *A. cerana cerana*. The distribution of hits for both species shows similar trends. Most hits fall within the low APM length, roughly between 20 and 60 amino acids. Additionally, the percentage of sequence identity displays a comparable distribution. The highest density of hits falls above 80% for the comparison obtained against APD3, and above 70% against DBAASP content. Putative AMPs with high sequence identity (above 90%) comprise the main fraction of identified sequences (Figure 2 and Figure 3; panels b, d). Such a pattern is a clear indication of very high conservation of the putative bee AMPs. It is worth noting that APD3 reports *A. mellifera* as a source of only 10 AMPs, and DBAASP of 18. In the case of *A. cerana, cerana*, the number of deposited sequences is even lower: one for APD3 and seven for DBAASP.

Analysis of the results reveals that the hit map can be simplified by grouping obtained hits by target protein. The single most abundant group are histone-related proteins that range from 100.0%, in the case of the *A. cerana cerana*/APD3 alignment pair, to 28.7% in the case of *A. mellifera*/DBAAMP. When eliminated, the total hits are reduced to 46.3%, 71.3%, 0.0%, and 34.3% for *A. mellifera* | APD3 and DBAAMP and *A. mellifera* | APD3 and DBAAMP respective alignment pairs. The remaining hits can be assigned to the following major groups: melittins, abaecins, defensins, apidaecins, glyceraldehyde-3-phosphate dehydrogenases, and reactive oxygen modulators (only *A. mellifera*/DBAAMP). Three of the analyzed pairs produced a single hit involving hymenoptaecin (UniProt ID: Q10416, A0A2A3EDE3).

### 2.3. Mapping of AMPs on the Proteomic Target

Since the introduction of the AlphaFold database, effective mapping of the determined sequence-based characteristics onto a three-dimensional model is possible. For all of the proteomic targets identified in this study, there are 3D models available. Based on the obtained hits, we have mapped identified AMPs onto respective models. Table 2 presents a sample of the results for the alignment pair from the *A. mellifera*/APD3 dataset. The presented example is representative of the obtained results. Many of the hits present a similar level of redundancy where multiple AMPs are mapped onto a single proteome target with various levels of sequence identity. Complete mapping results are available in the Appendix A.

Three-dimensional models of the defensin and hymenoptaecin proteome targets are presented in Figure 4. In presented cases, the AMP domains are firmly located in the C-terminal part of the protein. In both cases, the AMP domain is represented by a well-defined secondary structure: an α-helix bundle with a small β-sheet domain in the case of defensin, and an all-β-sheet domain in the case of hymenoptaecin. Both domains are also characterized by a high AlphaFold pLDDT score describing high confidence in the model correctness [40].

## 3. Discussion

The antimicrobial properties of various natural products are often well established on the basis of historical precedence and, more importantly, are confirmed by diligent scientific research. However, in many cases, the exact molecular mechanism behind the observed properties remains unclear. Similarly, understanding of the antimicrobial activity of honey is still being developed. The number of studies and publications on the mechanism of antimicrobial action in honey is constantly increasing [41]. We have briefly presented an overview of the known antimicrobial mechanisms and compounds of honey in the introduction to the current research paper. However, there are still knowledge gaps, due to the great variation in honey around the world.

The main factor determining the antimicrobial activity of flower honeys, except Manuka honey, was considered to be hydrogen peroxide (H_2_O_2_); however, the role of antimicrobial peptides of bee origin cannot be overlooked. It is commonly accepted that hydrogen peroxide is formed as a result of the conversion of glucose to gluconic acid, catalyzed by glucose oxidase (GOX) [6]. Recently, however, there have been new reports that have provided evidence for an additional nonenzymatic mechanism of hydrogen peroxide formation in honey, related to the presence of polyphenolic compounds. The role of honey polyphenols was emphasized primarily as the bioactive components of honey with the highest antioxidant potential. However, the pro-oxidant effect of these compounds has also been considered to be a result of catalysis by transition metals present in biological systems, i.e., Fe and Cu [42]. The pro-oxidant effect of polyphenols may explain the relationship, observed in many studies, between the presence of polyphenols in honey and their antibacterial activity—the most valuable and frequently studied biological activity of honey [2,3,10,43]. Research, presented by Bucekova et al. [44] on Slovak flower honeys, showed no correlation between the content of GOX and H_2_O_2_, and similar antibacterial activity (measured by the minimal inhibitory concentrations (MIC) value against *S. aureus* and *P. aeruginosa*) in honeys diluted rather than incubated, as well as honeys incubated with proteinase K (which inactivates GOX and other proteins and peptides, e.g., defensin-1). Similar results were obtained by Grecka et al. [14], who determined the overall antibacterial activity and H_2_O_2_ concentration in 144 honey samples from northern Poland. Although a significant correlation was documented between the level of accumulated H_2_O_2_ and the antibacterial activity of Polish honey samples, samples with low MIC values were characterized by low H_2_O_2_ concentrations. Thus, the presence of specific polyphenols is probably responsible for the differentiation of antibacterial activity between varieties, and between different samples of the same variety. On the other hand, the impact of honeybees on the antibacterial activity of honey seems to be more stable, which was the main hypothesis verified in the present study.

Due to the many factors shaping the antibacterial activity, direct comparisons of the antibacterial activity of honeys, produced by different species of bees, are difficult and do not allow one to separate the contribution of the bee itself and the substance collected in the environment. The varied availability of nectar sources, resulting from different living conditions of *A. mellifera* (the Western honeybee distributed all over the world) and *A. cerana* (the Asiatic honeybee) [45], means that the variety of plant substances involved masks the effect caused by bees’ active substances, including AMPs. The sequence similarity of antimicrobial peptide transcript genes, from Asiatic and Western honeybees, has previously been demonstrated [46], while the number of AMPs in the case of *A. cerana* was higher.

To understand protein-based antimicrobial attributes, we have assessed the properties of the proteome belonging to the selected bee species, *Apis mellifera* and *Apis cerana cerana*, through in silico comparative peptide composition and distribution analysis. We have focused our investigation on antimicrobial peptides (AMPs) and attempted to verify their presence and distribution in the published honeybee proteomes by applying established bioinformatic tools. The reference databases, containing collections of AMPs used in our investigation, were analyzed, and the degree of redundancy of the content was assessed. We have established that the collected data do indeed show a significant level of similarity when the content is analyzed with respect to the database itself, as well as in the comparison between two selected databases. For example, more than 1000 hits in APD3, against APD3 database content alignment, share 80% identity (Figure 1a, Appendix A). Similar amino acid sequence patterns were detected when the content of the DBAASP database was compared with APD3. It is worth noting that, in this case, the 100% peak in the hit frequency histogram symbolizes identical peptide sequences in both databases (Figure 1b, Appendix A).

In silico analyses were previously only used to analyze the interaction of the honeybee venom protein with the spike protein of the bioactivity of the Ebola virus [47]. In turn, for plants, the in silico and in vitro bioactivity was used to confirm antimicrobial properties of some rapeseed seed storage proteins by determining their similarity with other plant antimicrobial peptides through the conservation of sequence motifs and specific amino acids, as well as 3D structural analysis [48]. Thus, the approach we have proposed is unique and, to our knowledge, has not been previously used.

## 4. Materials and Methods

### 4.1. Data Sources

All the sequences used in the paper were downloaded on 23 January 2023 from the following sources:Antimicrobial Peptide Database (APD3, https://aps.unmc.edu/ accessed on 23 January 2023) [37];Database of Antimicrobial Activity and Structure of Peptides (DBAASP, https://dbaasp.org/ accessed on 23 January 2023) [38];UniProt proteome sequences (https://uniprot.org/ accessed on 23 January 2023) reference proteomes section (https://www.uniprot.org/proteomes accessed on 23 January 2023) [39].

### 4.2. Alignment Programs

The diamond (v. 2.0.15.153, [48]) software was obtained following the author’s instructions at https://github.com/bbuchfink/diamond accessed on 23 January 2023. The alignments were performed with default parameter settings using either APD3 as a generated reference database for the redundancy analysis or the respective proteome-based generated reference databases for the AMP mapping calculations. Complete alignment results are available in the Appendix A.

### 4.3. Data Visualization

Result analysis and visualization were performed in R (v. 4.2.3, [49]) using RStudio IDE (v. 2023.03.0, [50]). The plots were generated using the ggplot package.

The structural analysis and AMP mapping were performed in open source PyMOL (v. 2.5.0, [51]).

## 5. Conclusions

In the current paper, we have presented the results of the in silico analysis of the proteome of two honeybee species: *Apis mellifera* and *Apis cerana cerana*. We have focused our investigation on the comparative assessment of putative antimicrobial components. In our investigation we have identified the respective components through the sequence alignment procedure, utilizing two selected antimicrobial peptide sequence databases, APD3 and DBAASP. In silico studies have shown that antibacterial peptides are present and distributed in the published honeybee proteomes of both tested species. We have demonstrated the adequacy of the procedure on the selected examples, showing plausible matching results further confirmed by the structural analysis. Based on the obtained results, we have shown the possible flexibility of antimicrobial-sequence-encoded properties, based on the multiple AMP sequence being matched to the single proteome targets. The finding was further validated by mapping the peptide sequences on the 3D protein model. The structural analysis revealed that the AMP site is a single, well-defined domain with potentially conserved structural features. There were no significant interspecies differences in the composition of the antibacterial fraction of peptides. This may explain the observed variability between honey varieties, as the bees’ contribution to shaping the antibacterial activity of honey seems to be constant; the observed variability results mainly from nectar flow and/or other environmental factors. A new approach based on the use of advanced bioinformatics tools, to elucidate the mechanisms of antibacterial action, provides new information regarding the diverse biological activity of honey observed in vitro.

## Figures and Tables

**Figure 1 antibiotics-12-00830-f001:**
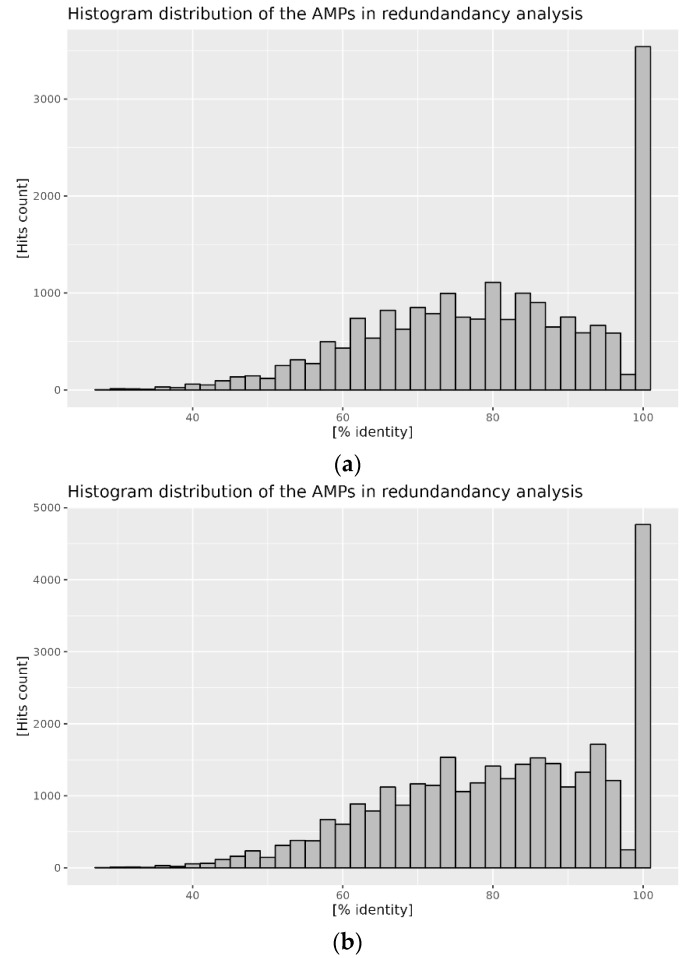
Histogram of the hits percent identity score: (**a**) content of the APD3 database compared to itself; (**b**) content of the DBAASP compared to APD3.

**Figure 2 antibiotics-12-00830-f002:**
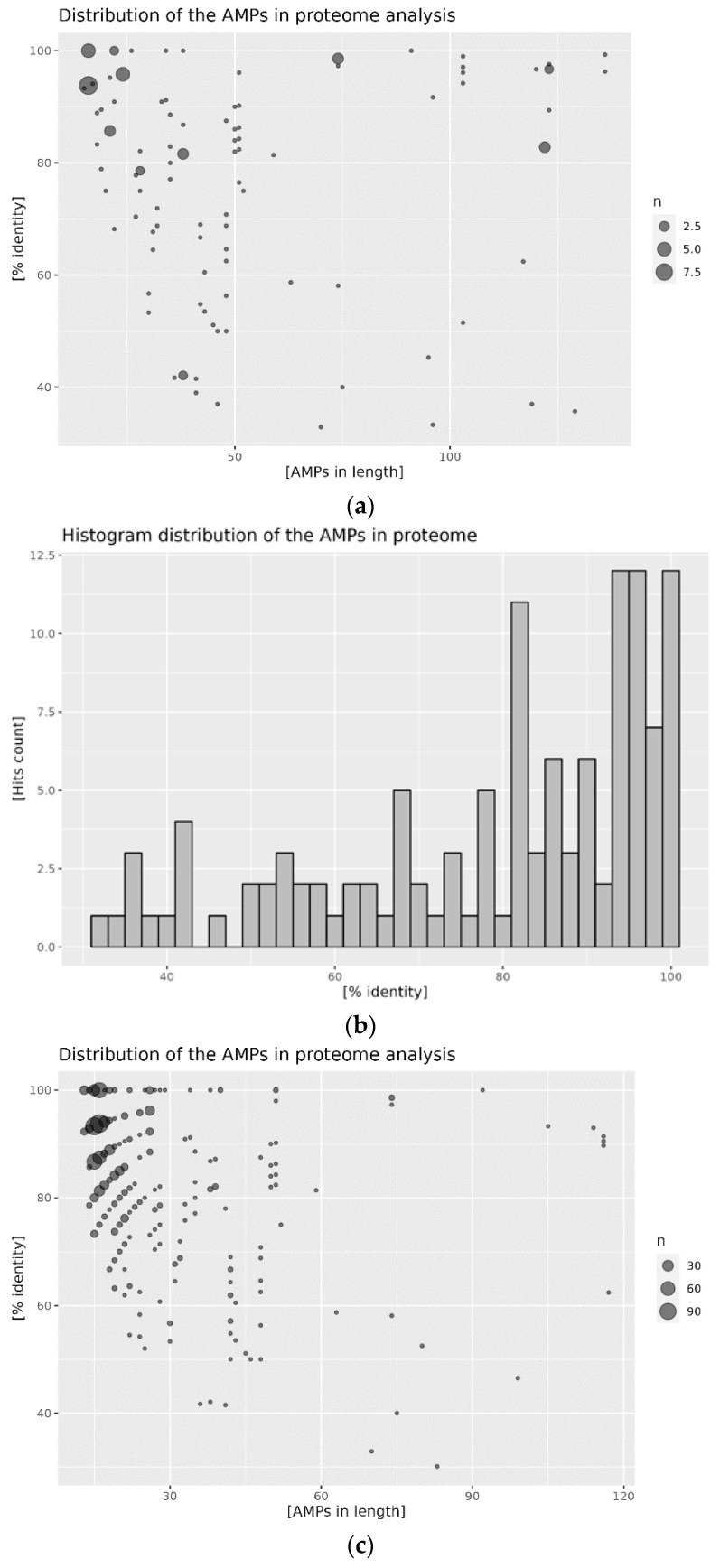
Distribution of AMPs in the proteome of *A. mellifera*: (**a**) hits distribution against the APD3 database, size of the point is proportional to the hit count; (**b**) histogram of the percentage identity of the hits against the APD3 database; (**c**) hits distribution against the DBAASP database, size of the point is proportional to the hit count; and (**d**) histogram of the percent identity of the hits against the DBAASP database.

**Figure 3 antibiotics-12-00830-f003:**
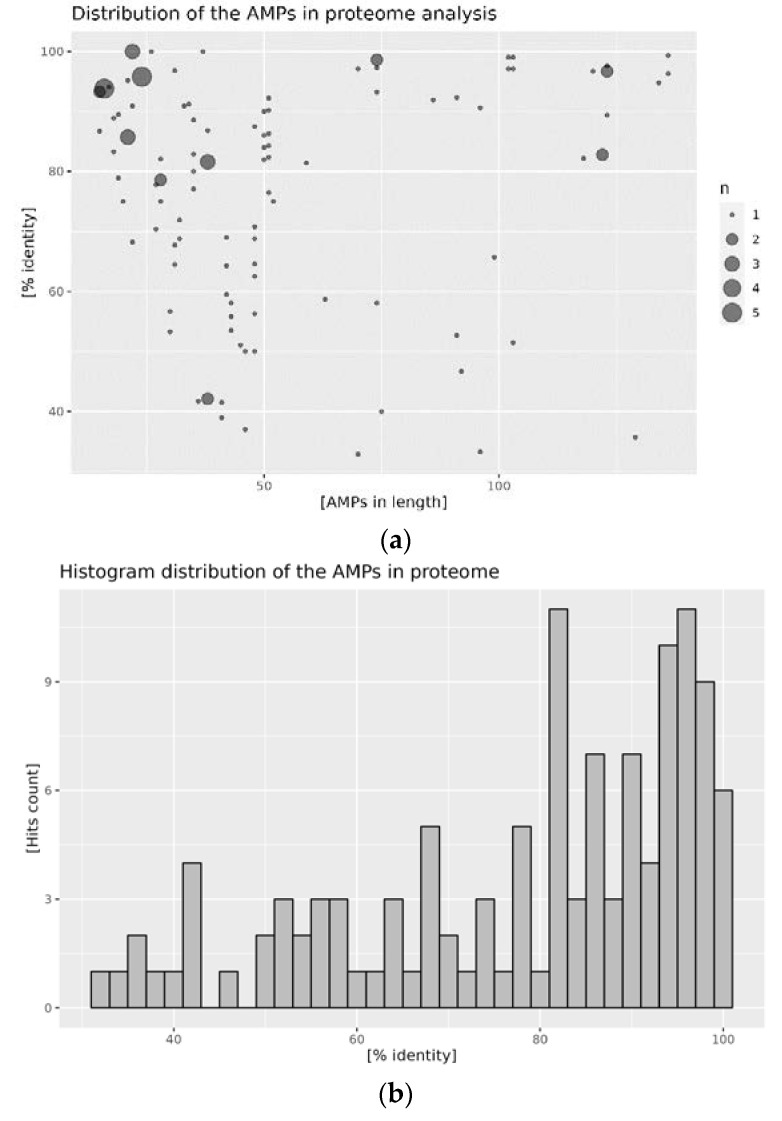
Distribution of AMPs in the proteome of *A. cerana cerana*: (**a**) hits distribution against the APD3 database, size of the point is proportional to the hit count; (**b**) histogram of the percentage identity of the hits against the APD3 database; (**c**) hits distribution against the DBAASP database, size of the point is proportional to the hit count; and (**d**) histogram of the percentage identity of the hits against the DBAASP database.

**Figure 4 antibiotics-12-00830-f004:**
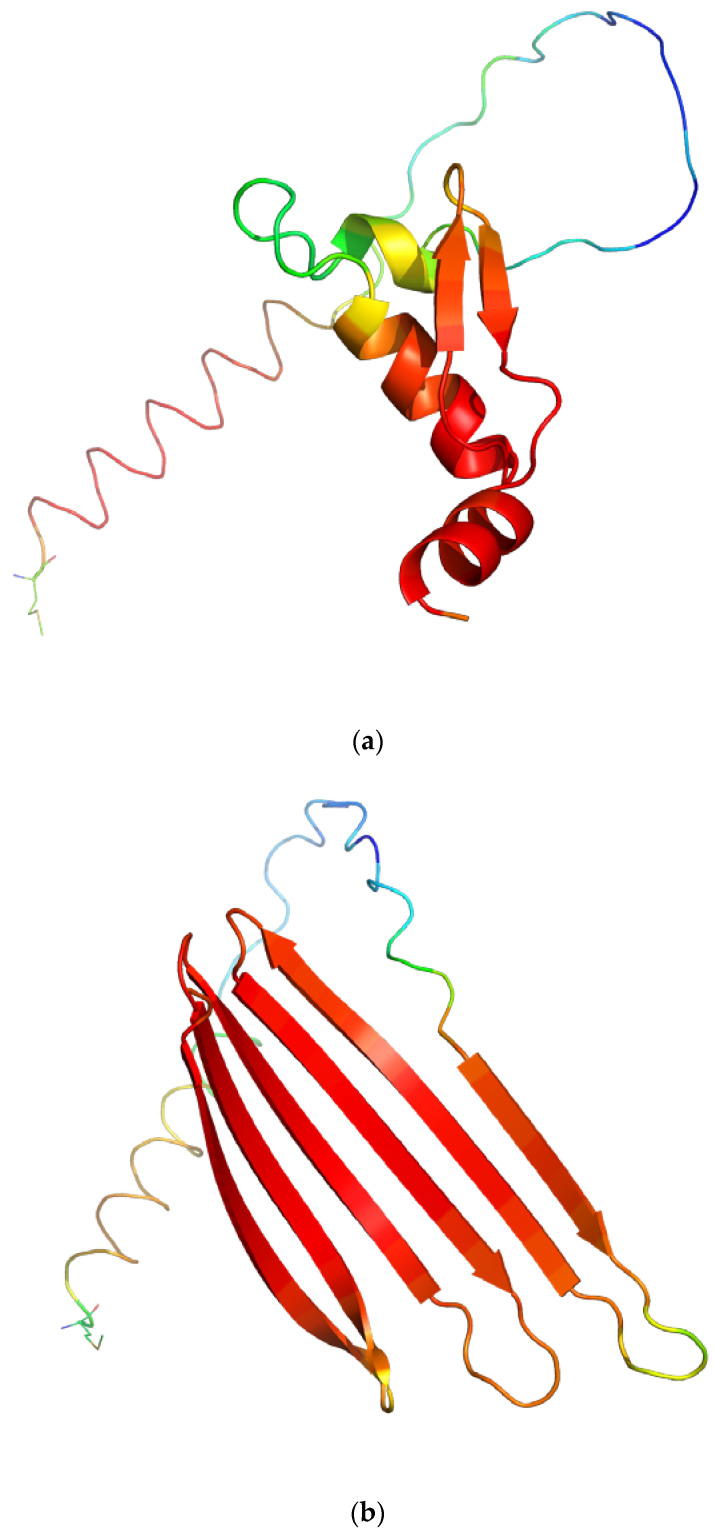
AlphaFold models for selected proteome targets based on hits from the alignment pair *A. mellifera* | APD3: (**a**) defensin (UniProt ID: P17722); (**b**) hymenoptaecin (UniProt ID: Q10416). The AMP encoding domain of the protein molecule is represented as a graphic; the N-terminal end of the target is indicated by the stick representation of the amino acid. Color coding represents confidence in the generated model (red: very high; blue: very low) according to the AlphaFold pLDDT score. Figure generated with the PyMOL program (v. 2.5.0, Schrödinger, New York, NY, USA). Respective coordinate data for structure visualization downloaded from https://alphafold.ebi.ac.uk/ accessed on 13 March 2023.

**Table 1 antibiotics-12-00830-t001:** Examples of antibacterial compounds in honey.

Honey Antimicrobial Component	Mechanism	Microorganisms	References
Sugars	Osmotic pressure	*Staphylococcus aureus* *Pseudomonas aeruginosa*	Abdel-Azim et al., 2019 [11] Proaño et al., 2021 [12]
Polyphenols	*Antioxidant activity* *Immunomodulation* H_2_O_2_ generation Inhibition of bacterial enzymes Membrane disruption Chelation of metal ions DNA/RNA/protein disorders	*Escherichia coli* *Bacillus subtilis* *Staphylococcus aureus* *Staphylococcus lentus* *Pseudomonas aeruginosa* *Klebsiella pneumoniae*	Estevinho et al., 2008 [13] Grecka et al., 2018 [14] Nolan et al., 2019 [15]
Methylgyoxal (MGO)	*Alterations in bacterial structure, limiting bacterial motility and adherence*	*Staphylococcus aureus Pseudomonas aeruginosa* *Staphylococcus epidermidis* *Klebsiella pneumoniae* Escherichia coli	Rabie et al., 2016 [16] Deng et al., 2018 [17] Girma et al., 2019 [18]
Glucose oxidase	H_2_O_2_ generation	*Escherichia coli* *Bacillus subtilis* *Pseudomonas aeruginosa*	Kwakman et al., 2011 [19] Bucekova et al., 2014 [6] Brudzynski and Sjaarda, 2015 [20]
Peptides (mainly defensin-1)	Immunomodulation, membrane disruption, and inhibition of bacterial cell wall synthesis	*Staphylococcus aureus* *Pseudomonas aeruginosa* *Bacillus subtilis* *Escherichia coli*	Kwakman et al., 2011 [19] Proaño et al., 2021 [12]

**Table 2 antibiotics-12-00830-t002:** Selected example of the alignment pair for the *A. mellifera*/APD3 analysis. Complete data available in Appendix A.

APD3 APMs	Proteome Target	% Identity	APD3 Sequence	Target Sequence
00226| Royalisin	sp|P17722| DEFI_APIME	96.1	VTCDLLSFKGQVNDSACAANCLSLGKAGGHCEKVGCICRKTSFKDLWDKRF	VTCDLLSFKGQVNDSACAANCLSLGKAGGHCEKGVCICRKTSFKDLWDKRF
02331| B.	sp|P17722| DEFI_APIME	76.5	VTCDLLSIKGVAEHSACAANCLSMGKAGGRCENGICLCRKTTFKELWDKRF	VTCDLLSFKGQVNDSACAANCLSLGKAGGHCEKGVCICRKTSFKDLWDKRF
01752| Defensin-NV	sp|P17722| DEFI_APIME	75.0	VTCELLMFGGVVGDSACAANCLSMGKAGGSCNGGLCDCRKTTFKELWDKRFG	VTCDLLSFKGQVNDSACAANCLSLGKAGGHCEKGVCICRKTSFKDLWDKRFG
01358| A.	sp|P17722| DEFI_APIME	60.5	VTCDLLSFEAKGFAANHSLCAAHCLAIGRRGGSCERGVCICRR	VTCDLLSFKGQVNDSACAANCLSLGKAGGHCEKGVCICRK
02735| Oryctes	sp|P17722| DEFI_APIME	53.5	LTCDLLSFEAKGFAANHSLCAAHCLAIGRKGGACQNGVCVCRR	VTCDLLSFKGQVNDSACAANCLSLGKAGGHCEKGVCICRK
01213| Hymenoptaecin	sp|Q10416| HYTA_APIME	100.0	RGSIVIQGTKEGKSRPSLDIDYKQRVYDKNGMTGDAYGGLNIRPGQPSRQHAGFEFGKEYKNGFIKGQSEVQRGPGGRLSPYFGINGGFRF	RGSIVIQGTKEGKSRPSLDIDYKQRVYDKNGMTGDAYGGLNIRPGQPSRQHAGFEFGKEYKNGFIKGQSEVQRGPGGRLSPYFGINGGFRF

## Data Availability

Publicly available datasets were analyzed in this study. This data can be found here: APD3, https://aps.unmc.edu/, DBAASP, https://dbaasp.org/, https://www.uniprot.org/proteomes/ (accessed on 23 January 2023) IDs: UP000005203, UP000242457. The data presented in this study are available as Appendix A.

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
