# Peer review of "Assessing the Antimicrobial Properties of Honey Protein Components through In Silico Comparative Peptide Composition and Distribution Analysis"

_antibiotics, 2023, doi:10.3390/antibiotics12050830_

Round 1

Reviewer 1 Report

The authors present a study including screening and identifying antimicrobial proteins in honey. Some specific comments that may elevate the overall quality and readability of the manuscript:
Introduction:
A Table with some antimicrobial properties+microorganism (for example, some polyphenols)need to be added since this is the central core of honey properties and the justification for this research. Moreover, a paragraph regarding the botanical and geographical diversification of honey, which reflects the antimicrobial and antioxidant properties, must also be added.
Please address some issues regarding the properties of the selected bee species and their honey products, propolis, royal jelly etc.
L50-52:There are numerous modes of action. A few references for honey antimicrobial activity against pathogens like E. coli and S. aureus would be excellent.
L148: "reviles", please use another word.
Discussion:
L190-192: I suggest you elaborate more and provide some recent citations. Antimicrobial and antioxidant modes of action of honey products are not a mystery, yet there are some gaps in terms of unknown substances. For example, dark-coloured honey offers more antioxidant capacity than honey originating from flowers.
L288-289: That is evidence of your research. Is that a general fact and rule? If the answer is yes, then please provide some references.
Some references that may help(NOT COMPULSORY TO USE)

Stefanis, C.; Stavropoulou, E.; Giorgi, E.; Voidarou, C.; Constantinidis, T.C.; Vrioni, G.; Tsakris, A. Honey’s Antioxidant and Antimicrobial Properties: A Bibliometric Study. Antioxidants 2023, 12, 414. https://doi.org/10.3390/antiox12020414

Danieli, P.P.; Lazzari, F. Honey Traceability and Authenticity. Review of Current Methods Most Used to Face this Problem.J. Apic.Sci.2022,66, 101–119. [CrossRef]

Valverde, S.; Ares, A.M.; Elmore, J.S.; Bernal, J. Recent trends in the analysis of honey constituents.Food Chem.2022,387, 132920.[CrossRef] [PubMed]

Iftikhar, A.;  Nausheen, R.;  Mukhtar, I.;  Iqbal, R.K.;  Raza, A.;  Yasin, A.;  Anwar, H. The regenerative potential of honey:  Acomprehensive literature review.J. Apic. Res.2022,62, 97–112. [CrossRef]

Asma, S.T.; Bobi ̧s, O.; Bonta, V.; Acaroz, U.; Shah, S.R.A.; Istanbullugil, F.R.; Arslan-Acaroz, D. General Nutritional Profile of BeeProducts and Their Potential Antiviral Properties against Mammalian Viruses.Nutrients2022,14, 3579. [CrossRef] [PubMed]

Wang, X.; Chen, Y.; Hu, Y.; Zhou, J.; Chen, L.; Lu, X. Systematic Review of the Characteristic Markers in Honey of VariousBotanical, Geographic, and Entomological Origins.ACS Food Sci. Technol.2022,2, 206–220. [CrossRef]

Feknous, N.; Boumendjel, M. Natural bioactive compounds of honey and their antimicrobial activity.Czech J. Food Sci.2022,40,163–178. [CrossRef]

Tumbarski, Y.; Topuzova, M.; Todorova, M. Food Industry Applications of Propolis: A Review.J. Hyg. Eng. Des.2022,40, 257–265.

Author Response

Answers to Reviewers comments:

Reviewer 1:

We highly appreciate Your efforts in improving our manuscript and thank you for each valuable comment which allow to clarify our scientific report.

The authors present a study including screening and identifying antimicrobial proteins in honey. Some specific comments that may elevate the overall quality and readability of the manuscript:

Introduction:

A Table with some antimicrobial properties+microorganism (for example, some polyphenols)need to be added since this is the central core of honey properties and the justification for this research. Moreover, a paragraph regarding the botanical and geographical diversification of honey, which reflects the antimicrobial and antioxidant properties, must also be added.

The Table summarizing main known antimicrobial compounds and mechanisms of honey was addeed. A paragraph regarding the botanical and geographical diversification of honey was included (lines 54-58).

Please address some issues regarding the properties of the selected bee species and their honey products, propolis, royal jelly etc.

Available informations regarding properties of bee products produced by different bee species were included (lins 59-64).

L50-52:There are numerous modes of action. A few references for honey antimicrobial activity against pathogens like E. coli and S. aureus would be excellent.

Some information about effect on antibiotic-resistant pathogens were added (lines 70-73).

L148: "reviles", please use another word.

The word was changed to „reveals“.

Discussion:

L190-192: I suggest you elaborate more and provide some recent citations. Antimicrobial and antioxidant modes of action of honey products are not a mystery, yet there are some gaps in terms of unknown substances. For example, dark-coloured honey offers more antioxidant capacity than honey originating from flowers.

The paragraph was rewritten, some references were included. In fact, the mechanism are not a mystery, but it is still being clarified. The information about differences about dark and light honeys was included in Introduction section.

L288-289: That is evidence of your research. Is that a general fact and rule? If the answer is yes, then please provide some references.

The Conlusion was rewitten, we put some new suggestion which should be verified experimentally and not yet reported in literature.

Some references that may help (NOT COMPULSORY TO USE)

Thank you for listed references, we used some of them.

Reviewer 2 Report

The manuscript can be suitable for publication after minor revision. My specific comments are given below:

Line 24: AMP- use the full form here.

Line 28: Keywords-- arrange alphabetically. Honey bee or honeybee - use uniformly throughout the manuscript.

Figure 2. Heading: proteom- check

Figure 3. proteom- check

Line 195: hydrogen peroxide (H2O2)

Line 259: proteom-- check

Author Response

Answers to Reviewers comments:

Reviewer 2:

We highly appreciate Your efforts in improving our manuscript and thank you for each valuable comment which allow to clarify our scientific report.

The manuscript can be suitable for publication after minor revision. My specific comments are given below:

Line 24: AMP- use the full form here.

The abbreviation was explained.

Line 28: Keywords-- arrange alphabetically. Honey bee or honeybee - use uniformly throughout the manuscript.

Keywords were arranged alphabetically.

Figure 2. Heading: proteom- check

Figure corrected accordingly.

Figure 3. proteom- check

Figure corrected accordingly.

Line 195: hydrogen peroxide (H2O2)

Was modified.

Line 259: proteom—check

The word was corrected.

Reviewer 3 Report

The manuscript on honey peptides and their antimicrobial properties is interesting and well written. The background is adequate and the methodology is scientifically sound. Good Discussion and many relevant references of good quality. The topic is original and very interesting for the natural products community. Some minor corrections:

1. Line 23: Mention the % sequence similarity values

2. Line 25: Briefly describe the conserved domain

3. Lines 60-63: Provide references for supporting these statements

4. Figure 4: 'Proteome' not 'proteom'

5. Lines 170-176: Correct font size. In 176, there are two full stops.

6. Line 186: 'antimcrobial' is in a different font size. Please correct all such inconsistencies in the text.

7. Line 215: minimal inhibitory concentrations (MIC)  

Author Response

Answers to Reviewers comments:

Reviewer 3:

We highly appreciate Your efforts in improving our manuscript and thank you for each valuable comment which allow to clarify our scientific report.

The manuscript on honey peptides and their antimicrobial properties is interesting and well written. The background is adequate and the methodology is scientifically sound. Good Discussion and many relevant references of good quality. The topic is original and very interesting for the natural products community. Some minor corrections:

1. Line 23: Mention the % sequence similarity values

Respective information included.

2. Line 25: Briefly describe the conserved domain

Brief description of stuctural features included.

3. Lines 60-63: Provide references for supporting these statements

Suitable references [26, 29] are now correctly linked to paragraph.

4. Figure 4: 'Proteome' not 'proteom'

Figure corrected accordingly.

5. Lines 170-176: Correct font size. In 176, there are two full stops.

The font size was unified in whole manuscript.

6. Line 186: 'antimcrobial' is in a different font size. Please correct all such inconsistencies in the text.

The font size was unified in whole manuscript.

7. Line 215: minimal inhibitory concentrations (MIC)

Was modified.
